Inter-annual variability influences the eco-evolutionary dynamics of range-shifting

Henry Roslyn C. 1 r01rch12@abdn.ac.uk
Bocedi Greta 1
Dytham Calvin 2
Travis Justin M.J. 1
1 Institute of Biological Sciences, University of Aberdeen , Aberdeen , UK
2 Department of Biology, University of York , Heslington, York , UK
White Ethan
Electronic publication date: 2014 Jan 2
Publication date: 2014
Volume: 2
Electronic Location ID: e228
Received 2013 Oct 21; Accepted 2013 Nov 30
Copyright: © 2013 Henry et al.
Copyright year: 2013
Copyright holder: Henry et al.
License: This is an open access article distributed under the terms of the Creative Commons Attribution License, which permits unrestricted use, distribution, and reproduction in any medium, provided the original author and source are credited.
License URL: https://creativecommons.org/licenses/by/3.0/

Keywords: Dispersal, Climate change, RangeShifter, Environmental noise

Funding: Natural Environment Research Council, UK (NERC) NE/J008001/1 Funding from the Natural Environment Research Council, UK (NERC) supported this research. The funder had no role in study design, data collection and analysis, decision to publish, or preparation of the manuscript.

==============================
Understanding the eco-evolutionary dynamics of species under rapid climate change is vital for both accurate forecasting of biodiversity responses and for developing effective management strategies. Using an individual-based model we demonstrate that the presence and form (colour) of inter-annual variability in environmental conditions can impact the evolution of dispersal during range shifts. Under stable climate, temporal variability typically results in higher dispersal. However, at expanding margins, inter-annual variability actually inhibits the evolution of higher emigration propensities by disrupting the spatial sorting and natural selection processes. These results emphasize the need for future theoretical studies, as well as predictive modelling, to account for the potential impacts of inter-annual variability.

Introduction

Growing evidence suggests climate change is intensifying the inter-annual variability of the climate and the frequency of extreme weather events, further increasing the temporal correlation of unusual conditions (Coumou & Rahmstorf, 2012; Hansen, Sato & Ruedy, 2012). To survive global warming species must either adapt in situ or shift their ranges to areas of newly suitable habitat (Holt, 2003; Parmesan, 2006). Species have already exhibited distributional changes that are consistent with those predicted by climate impact modelling (Chen et al., 2011). A species’ dispersal ability is central for determining its range shifting potential and is thus a key parameter in predictive models. While in most models of range expansion individuals disperse with a constant probability and/or move a distance drawn from a constant dispersal kernel (Kot, Lewis & Van Den Driessche, 1996), there is increasing evidence from both theoretical and empirical studies that dispersal will come under strong and rapid selection during range shifts (Travis & Dytham, 2002; Hughes, Dytham & Hill, 2007; Phillips, Brown & Shine, 2010a; Phillips, Brown & Shine, 2010b; Phillips et al., 2010; Kubisch et al., 2013). Although we understand how dispersal should evolve in a stationary range with inter-annual variability in climate conditions (Kun & Scheuring, 2006; Bocedi, Heinonen & Travis, 2012), theory on range shifting responses to climate change have typically only modelled changes in the mean rate of climate (Boeye et al., 2013; Henry, Bocedi & Travis, 2013). Thus most have not considered the additional effect of inter-annual variability, despite the recognition that most natural systems exist in environments that are variable and positively correlated (Inchausti & Halley, 2002; Vasseur & Yodzis, 2004).

In temporally constant and stable environments there is typically only limited selection driving dispersal, e.g., pressure from kin selection (Ronce, Gandon & Rousset, 2000) and inbreeding avoidance (Perrin & Mazalov, 1999; Perrin & Mazalov, 2000), and if emigrants suffer a cost by leaving their natal patch, dispersal will remain low (Travis & Dytham, 1999; Dytham, 2009). Environmental variability, however, leads to greater uncertainty, increasing spatio-temporal variability of environmental conditions and population density. In these situations a bet-hedging strategy, dispersing to escape the possibility of local disaster, becomes advantageous (Friedenberg, 2003). Thus several previous studies have demonstrated selection for increased dispersal when environmental variability is high and uncorrelated (Kun & Scheuring, 2006; Bocedi, Heinonen & Travis, 2012). These results are from stationary ranges and the consequences of environmental variability for dispersal evolution during range shifting remain unknown. Here, we develop some initial theory to address this and show that environmental noise impedes the evolution of dispersal during range shifting.

Methods

For the purpose of this study we use a spatially-explicit individual-based modelling platform, RangeShifter (Bocedi et al., in press). We model a haploid species with asexual reproduction and discrete, non-overlapping generations on discrete random landscapes, where each individual is characterized by its position on the landscape and dispersal characteristics.

Population Dynamics

The landscape cells contain sub-populations of individuals, characterized by density-dependent dynamics with demographic stochasticity. After dispersing, an individual settling in a suitable cell within the climatic window (i.e., where K > 0) reproduces. The dynamics of local populations are simulated using an individual-based formulation of the Maynard Smith and Slatkin’s (1973) single species population model. In every generation (t), each individual produces a number of offspring randomly drawn from a Poisson distribution with mean μt given by the following formula: μt=r⋅Nt1+r−1⋅NtKb

where r is the maximum growth rate at low densities, b specifies the type of competition and here was assumed to be equal to 1 (‘contest’ competition). Nt is the number of individuals in a cell at time t, and K is the carrying capacity of the cell, both remain constant during offspring production. This method of determining the number of offspring introduces demographic stochasticity into the model. Individual emigration probabilities are initially randomly drawn from a uniform distribution between 0 and 1. Offspring inherit the same emigration probability as their parent with a probability of mutation (m) equal to 0.001. If a mutation occurs, the emigration probability of the offspring assumes a random value drawn from a uniform distribution between −0.1 and 0.1 + the previous trait value. After reproduction all the adult individuals die and the new populations consist of the offspring.

Dispersal

Emigration is density independent, at the start of each generation each new individual has a probability of emigrating determined by its emigration trait value. The distance each individual disperses is sampled from a negative exponential distribution with mean equal to 200 m (equal to two cells), while the direction is drawn from a uniform circular distribution (between 0 and 2π radians). Individuals settle in a patch if it is suitable; if the patch is unsuitable the individual dies. When all the dispersing individuals have either died or settled, the dispersal phase is concluded and the model continues to the next generation step. Note that in using this method not all individuals that have the opportunity to emigrate do effectively emigrate. A proportion will draw a direction and distance resulting in them landing and remaining in their natal cell. However, as the mean dispersal distance is constant, this does not lead to confounding effects between evolution of emigration probability and dispersal distance.

Landscape

The simulations are performed on discrete random landscapes of 2000 by 50 cells with a single habitat type. The resolution of each cell is 100 m. Reflective boundary conditions are applied. The landscape was randomly composed of suitable and unsuitable cells and for this experiment 30% of the cells are randomly selected to be suitable. On top of this random landscape an environmental gradient in cell carrying capacity (K) is then applied linearly in space along the x axis. The gradient method follows that of Travis & Dytham (2004), optimal conditions (Kopt = 100) decline linearly to conditions that do not allow population survival (K = 0) resulting in a climate window of suitable habitat space. The carrying capacity of a suitable cell with coordinates x, y (K(x,y)) is set by the following equation: Kx,y=Kopt−|x−xopt|G

where |x−xopt| is the distance from the cell to the optimum and G is the gradient steepness (G = 2). Without environmental stochasticity this method results in a climate window that is 100 cells wide in the x dimension of the landscape. K is therefore constrained to be greater than or equal to zero.

To simulate a period of climate change the climatic window moves unidirectionally from left to right (low to high values) along the x axis at speeds varying from 0.125 to 2.0 x co-ordinates ⋅ t−1.

Environmental noise

To simulate inter-annual variability, a time series of K modifiers is generated using a widely used first-order autoregressive process detailed by Ruokolainen et al. (2009). Carrying capacity is varied temporally according to the following equation: Kx,y,t=Kx,y,0+K⋅εt

Here, Kx,y,t is the carrying capacity of the cell with coordinates x and y at time t, Kx,y,0 is the expected value of K for the cell in absence of stochasticity and K is the carrying capacity in absence of gradient and stochasticity (i.e., Kopt). εt is the time series value determined by εt=κεt−1−1+ωt−11−κ2

where ω is a random normal variable sampled from N(0, σ) and κ is the autocorrelation. Here we set standard deviation (σ) to 0.25. This equation produces positively correlated (red) noise when κ is greater than zero and white noise when κ is equal to zero.

Simulation experiments

In each simulation, cells are initialised at carrying capacity and the optimum x co-ordinate is held constant for 500 generations, shifted for 300, then stable for 500. Individual emigration probability (d) is randomly drawn from a uniform distribution between 0 and 1. We run simulations with no noise, uncorrelated (white) noise (standard deviation σ = 0.25, autocorrelation κ = 0.0) and positively correlated (red) noise (σ = 0.25, κ = 0.9) to explore how the presence and form of environmental variability influences the evolution of dispersal during range expansion for 10 different rates of climate change (between 0 to 2 x co-ordinates ⋅ t−1).

Results

Prior to climate change the mean emigration probability is low in the core and increases towards the range margins (Fig. 1A). The gradient in emigration probability becomes more pronounced with environmental noise, with the highest emigration rates at the margins evolving under white noise. Under red noise, populations are also surviving (at least some of the time) further from the core. Consistently, across all scenarios during climate change (Fig. 1B), the emigration probability across the whole range increases but the pattern of the dispersal gradient across the range changes, with low emigration at the rear of the range and higher emigration at the front. When temporal environmental variability is present, at the rear of the range the evolved emigration rate remains as before; white noise resulted in higher dispersal than red or no noise. However, interestingly, at the expanding margin a switch occurs and scenarios with no noise evolve the highest emigration rates. Apart from for the slowest rates of climate change we find the same pattern, the mean emigration probability at the range front is higher in scenarios with no environmental noise (Fig. 2). The difference between the emigration probabilities evolving under the different conditions decreases for the highest rates of climate change.

Figure 1 Mean emigration probability across the species range for different environmental noise scenarios.

(A) Prior to climate change, generation 500 and (B) following 300 generations of climate change, generation 800. Black, grey and red points represent the scenarios with no, white and red environmental noise respectively. The data shown are the averages of 2000 replicates for each scenario, bars represent the standard error. Points with no standard error bars at the margins of the range are generated from just one simulation run where a single simulation has produced a slightly wider range than usual.

Figure 2 Mean and standard error of emigration probability of individuals in the front-most five rows over the last 30 generations for different rates of climate change.

Black, grey and red points represent the scenarios with no, white and red environmental noise respectively. The data shown are the averages of the first 100 replicates for each scenario.

Discussion

Before climate change, emigration probability across the range evolves in response to both the gradient structure and the nature of environmental noise. With a gradient in carrying capacity, spatiotemporal variance in population dynamics increases towards the margin where K is lower and demographic stochasticity higher. This favours higher dispersal at the margins than in the core due to bet hedging (Ronce, 2007; Kubisch, Hovestadt & Poethke, 2010). Thus, in all scenarios, we find higher dispersal at the margins compared to the core. Environmental variability, particularly white noise, no temporal autocorrelation, further increases the temporal variance and its effects are greatest at the already fragile range margins. The gradient in emigration probability is therefore more pronounced with substantially higher dispersal at the margins. Similar to Mustin et al. (2013), we also find the range extent can become larger in positively correlated, red noise, scenarios as consecutive generations of good conditions allow populations to colonize marginal habitats that would otherwise remain unoccupied.

In this study our focus lies on the expanding margin however at this point it is also important to highlight that there is considerable scope for work exploring the eco-evolutionary dynamics at retreating margins. During climate change the emigration rate rapidly evolves to higher levels at the front of the range. This evolutionary process is well documented (Hughes, Dytham & Hill, 2007; Phillips, Brown & Shine, 2010a; Phillips, Brown & Shine, 2010b; Phillips et al., 2010) and is the result of spatial sorting and natural selection (Shine et al., 2011). This increase in dispersal enables species to better track areas of suitable habitat as climate change progresses, and may effectively act as an evolutionary rescue; without this process species might sometimes become extinct (Boeye et al., 2013; Henry, Bocedi & Travis, 2013). Furthermore, irrespective of noise presence or colour, as the rate of climate change increases, the mean emigration probability of individuals also increases. This is consistent with previous results that higher rates of climate change drive the evolution of higher rates of dispersal (Boeye et al., 2013). In all scenarios, at the end of climate change, the emigration probability is much lower at the trailing edge of the range compared to the front and environmental noise typically increases selection for higher emigration rates. However, at the front there is an interesting switch and now, rather than temporal environmental variability increasing dispersal, it is in fact disrupting its evolution. The environmental gradient results in poorer conditions (lower K) at the range front compared with the core, these patches are therefore more susceptible to extinction in bad years. Under temporal variability there is no longer a smooth advance of suitable environmental space, instead populations will expand somewhat only to be knocked back following a run of poorer years, this effect being more pronounced under positively autocorrelated variability. The most dispersive individuals assorted into the furthest advanced patches in a period of better conditions will be those prone to extinction when a sequence of poorer years occurs. This effectively disrupts the spatial sorting and natural selection processes, reducing the evolved dispersal at the front. Once climate change has ended, the dispersiveness of individuals evolves back to levels similar to those prior to climate change. Although our choice of landscape should not quantitatively change the results, we highlight that it may influence the underlying evolutionary processes after climate change. For example, with narrower climate windows the dispersiveness of the whole population will increase rapidly as all individuals are forced to track a narrower area of habitat and thus less dispersive genotypes will be lost. After climate change the decrease in dispersiveness back to levels similar to those prior to climate change therefore requires the evolution of lower dispersive genotypes to replace those lost in the range shifting process. However with increasingly wider climate windows and invasion scenarios, where individuals are initialised at one end of the landscape and allowed to expand across empty space, habitat at the rear of the range is less likely to disappear. Thus less dispersive genotypes at the rear of the range are unlikely to be lost and once climate change has ceased the return to lower average dispersal will not rely solely on new mutations but can be achieved by the spread of existing genotypes.

We suggest that similar effects of temporal environmental variability may occur for other life-history traits that are likely to come under selection during range expansion (Burton, Phillips & Travis, 2010; Phillips, Brown & Shine, 2010a; Phillips, Brown & Shine, 2010b; Phillips et al., 2010). These effects may, in turn, disrupt the process of range shifting as failure to evolve life history traits that promote range expansion may render species less able to track shifting climate. Mustin et al. (2013) have already demonstrated that positively correlated temporal variability increases the extinction risk in species during a period of climate change, particularly when climate change is rapid. Mustin et al. (2013) did not include evolution in their model and the impact of environmental variability is attributed to poor runs of environmental conditions increasing patch extinction rates towards the front therefore reducing the pool of patches from which the next wave of expansion can occur. Our results highlight another potential mechanism whereby noise could increase extinction risk through the disruption of dispersal evolution; in this case increasingly correlated environmental variability acts to constrain the upwards evolution of dispersal at the expanding front thus reducing the potential for dispersal evolution to rescue a species from climate change. We realise that the results presented here are sensitive to mutation rate with the evolution of higher dispersal occurring faster with higher rates of mutation and therefore the effect of interannual variability is reduced. However, the value used in the results presented is already relatively high for such a simulation of a quantitative trait so we believe the general effect is likely too robust for much realistic parameter space. Of course, future modelling efforts should strive to establish a suite of models with explicit and realistic genetic architecture.

Here, we have used a simple single species model and have explored scenarios where the population dynamics are characterized by contest competition. It is well understood that as a species’ dynamics become more complex through over-compensatory density dependence, dispersal is typically selected upwards (Holt & Mcpeek, 1996). However, it is not clear how these inherent complexities of a population’s dynamics will interact with environmental variability in driving the eco-evolutionary dynamics of range expansion, this is a topic worthy of future work.

Additionally, some recent studies have begun to explore how dispersal evolves in communities (e.g., Schreiber & Saltzman, 2009; Pillai, Gonzalez & Loreau, 2012; Travis et al., 2013) and there is concurrent interest in asking how communities will respond to rapid climate change (Norberg et al., 2012; Urban, Tewksbury & Sheldon, 2012; Singer, Travis & Johst, 2013; Phillips, Brown & Shine, 2010a; Phillips, Brown & Shine, 2010b; Phillips et al., 2010). Given that there is already good theoretical evidence that environmental noise can be filtered in non-intuitive ways by the dynamics of interacting species (Ruokolainen, Fowler & Ranta, 2007; Ranta et al., 2008), it will also be important to investigate the interplay between species interaction and dispersal evolution during range shifts that occur in temporally variable environments.

The nature of the population dynamics and the complexities of, for example, stage structure, dispersal stages, environmental dependent transition rates, will almost certainly influence how life histories evolve during expansions as a function of interannual variability. It is beyond the scope of this paper to encompass all possible complexity. However our preliminary study highlights the importance of inter-annual variability on climate induced range shifts and our results emphasize that theoretical studies investigating the impact of climate change should certainly consider the effects of environmental noise. Failure to do so risks reaching erroneous conclusions about the evolution of dispersal and overestimation of species ability to tolerate a period of climate change. This also holds true in an applied setting. European and North American systems are subject to decadal cycles of variability due to the Atlantic Multidecadal Oscillation (Sutton & Dong, 2012) greatly influencing species demography. For example, a drought lasting several years would impact upon recruitment and population persistence of many plant species. However, most current projections of climate impact are based on trends in mean climate and do not capture this potentially crucial effect. Future modelling, whether statistical or process-based, should urgently seek to incorporate inter-annual variability.

Additional Information and Declarations

Competing Interests

Author Contributions

The authors declare they have no competing interests.

Roslyn C. Henry conceived and designed the experiments, performed the experiments, analyzed the data, wrote the paper.

Greta Bocedi conceived and designed the experiments, contributed reagents/materials/analysis tools, wrote the paper.

Calvin Dytham conceived and designed the experiments.

Justin M.J. Travis conceived and designed the experiments, contributed reagents/materials/analysis tools.

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
