# Peer review of "Inter-annual variability influences the eco-evolutionary dynamics of range-shifting"

_PeerJ, doi:10.7717/peerj.228_

## Round 0.1 · original submission · Major Revisions

The reviewers and I agree that this is a solid paper. The reviewers make a number of valuable suggestions aimed at improving the paper and I encourage the authors to consider all of them carefully. There are two particularly important issues that need to be addressed.

1. Istvan Scheuring and Adam Kun both note a number of issues with respect to insufficient detail in the methods that need to be addressed. In particular Dr. Scheuring notes that one of the parameter values that is not reported has the potential to impact the interpretation of results depending on its value.

2. As noted by David Alonso the code underlying these simulations is not currently publicly available. Since the entire study relies on this code I consider this to be an important issue, as does PeerJ's policy (https://peerj.com/about/policies-and-procedures/#data-materials-sharing). I understand from the cover letter that this code is currently in review elsewhere. Please provide additional details about its current status and the potential to be able to properly cite it in this manuscript.

·

Basic reporting

No Comments

Experimental design

No Comments

Validity of the findings

So far I have always found the works of Dytham and Travis (and their coworkers) interesting and well written. This manuscript is no exception. Here they employ established methods to answer a novel question. The effect of climate on the behaviour of organism is a topic requiring special attention nowadays.

The authors emphasize the changes in the front of the shifting zone, but I find the change here less dramatic. In the scenarios with environmental noise dispersal rate increased by about 10%. The increase is only dramatic if there is no noise at all (black dots). On the other hand, at the trailing end of the zone dispersal rates lowered by at least by 20%. This I find rather strange, as here individuals should disperse as otherwise they will find themselves in an environment not suitable for them. Of course evolution does not have foresight, and thus many individuals might die at this part of the zone. Actually it would be interesting to see the number of individuals that die because they disperse to places that are not suitable for them. It would be a crude approximation for the load climate change burdens a population.

In every theoretical paper there are parameters that are set, and their values are not varied during the simulations. I do not suggest that more simulations to be carried out, but at least their potential effect should be discussed. For example, here the authors assumed scramble competition. What would happen if competition is more contest like? The severity of climatic change is not only the speed at which the optimal habitat shifts, but also the suitability of the habitats after the climate has changed. Here the habitable window of sites was rather narrow (100 cells). The slowest change shift the optimum by 37.5 cells, the fastest by 600. Actually, when the speed is higher than 1/6 then by the end of the 300 time steps the former optimum lies outside of the new window of habitable sites. What would happen if this window would be larger (i.e. G lower than 2, like 1 or 0.5)?

Additional comments

The phrase „reddening” is probably understood only by a fraction of the potential readership. I know noise type is labelled as being red in some landscape ecological article, so potentially it will be understood by those who are familiar with the field, on the other hand I find positive correlation (line 20) a much more understandable phrase to describe what is going on. I strongly recommend replacing reddening in this article.
L38 are you sure it is kin selection and not kin competition which you refer to here?
Formula after L62: The parameter b has not been explained in the text. Furthermore, please replace the asterisk with a multiplication dot or multiplication sign.
L74 what does it mean to have a mean dispersal distance of 200m?
L84 Please clarify the meaning of rows and x axis in the definition of the landscape. Climte changes in the larger dimension, which has 2000 cells. This is referred to as rows (methods section), but also as x coordinates (figures). X axis is more like columns rather than rows.
L88 the opt subscript in Kopt should not be italic as it is not a variable. Please correct this throughout.
L93 the first sentence here was already mentioned in the previous paragraph. On the other hand it would be nice to state that the resulting landscape is such, that without noise it has a 100 cells window in which populations are viable (because G=2)
L96 asterisk should be replaced
L97 I found the definition of environmental noise hard to follow. The variable K should not be introduced as it is Kopt. Kx,y,0 is actually K(x,y) from line 92. The indication of 0th time step here can actually cause misunderstanding, as K(x,y) changes through time (after the first 500 steps for 300 steps). And please replace the asterisk in line 101. Line 105 reads strangely. Should it be “Where omega is a random number sampled from a normal distribution, N(0,sigma)” It is not a standard normal distribution as that has sigma = 1.
L115 Replace the asterisk
L132, Maybe there is a comma missing after “climate change”.
L262. Stability should start with small s.
Figure 1: Minor tick marks should be added. The X axis label should also start with a capitalized letter as the Y axis label. This may be a personal preference, but I do not like error bars when they indicate errors smaller than the size of the symbol. Furthermore at places where the error is larger, they are hardly visible because they overlap (that cannot be helped here too much). And there are some point where there is no error bar. Why?
Figure 2 caption should refer to fig 1 or repeat the colour coding of the symbols.
Fig. 2: I hope there is a vector format version of this figure, as the one we got has a very low resolution. For example, i and l looks exactly the same in the labels. The X axis label speed should start with capital S. You might also consider replacing it with a more informative label that explicitly states that this is the speed of shifting climatic optimum. Here error bars would not overlap, and while they are generally small, there are some instances (in the left part) where they are important. You might consider enlarging the symbols a bit, and using open symbols instead of solid ones, so that the error bars are more visible.

·

Basic reporting

No Comments

Experimental design

No Comments

Validity of the findings

No Comments

Additional comments

This ms shows that the interannual signature of environmental variabiltiy can impact the evolution of dispersal during range shifts. The text reads well and results are conclusive.

All work is based on a particular individual based model (IBM). One can always ask: "To what extend results depend on the modeling approach? How can we validate conclusions from a particular computer simulation code without having access to the code itself? How do discrete vs non-discrete, and overlaping vs non-overlapping generations influnce results?". If any, this is the weakness and the main limitation of authors' contribution. They are aware of that and end the paper, in the last paragraph, by saying that their study highlights the importance of inter-annual variability on climate change induced range shifts. The ms can be seen as a preliminary result in this endeavor. However, I am convinced that their ms contributes to create this state of awareness. This is why, I believe, this ms deserves attention and is worth publlishing.

·

Basic reporting

The topic is interesting and the results are new and non-trivial. So generally I support the publication of the paper, but I have found numerous problematic points, mainly in the Methods section. Before publication these points should be addressed.


1. I don't think that title is informative enough, it is more general than the paper itself. I suggest a new title on the basis of present running title.

2. The description of methods is not sufficient. There are several points which are not clear enough for the reader (at least for me), some important points are missing and some are incorrect:
a) It is not completely clear from the text that $N_t$ doesn't change while new offsprings are produced. (line 62)
b) The value of parameter $b$ is not given and their role is not described. (line 62)
c) The value of parameter $m$ is not given (line 67). This could be crucial since all of the results remain adequate if $m$ is not unrealistically high for the effects.
d) It follows from point c) that authors should study the results in function of $m$.
e) What does 200m mean in this context?(line 74)
f) What is the reason authors use negative exponential distribution for dispersal kernel? (line 74)
g) It is not clear for me what is the geometry of the landscape? (subsection Landscape). It seems to me for example that while the gradient is on direction y in the text it is shown in direction x on the figures. A schematic figure would be helpful.
h) The sentence on line 86 is completely unclear for me? Is it true only initially? Why do they assume this?
i) I suspect that eq. at line 92 is valid only if K_{x,y}>=0, otherwise K_{x,y}=0.
j) In which direction does the climatic window move, from low to high y coordinate or vice verse? (line 96)
k) Eq. after line 104 is incorrect. It might be only a typo (a parenthesis is missing), but I emphasize that Ruokolainen et al used a different equation for generating $\varepsilon_t$. What is the connection between the two form?
l) The sentence at line 110 suggest that $K$ is selected randomly from [0 1], which is hardly the case.

Experimental design

The description of numerical experiment (as I indicated above) is not sufficiently described.

Validity of the findings

Point 2c listed above is connected with the validity.
If authors study the effect of white and red noise on dispersal evolution. Why they didn't do it for blue noise as well? I know that red noise is more adequate assumption, but by including blue noise they could demonstrate the trends more clearly.
I suggest to explain more clearly what does the statement on line 174 and 175 mean?

---

## Round 0.2 · accepted · Accept

Thank you for carefully considering and addressing my comments and those of the reviewers. Congratulations on the acceptance of your software paper at Methods in Ecology and Evolution.